# ROBUST INSTRUCTION-FOLLOWING IN A SITUATED AGENT VIA TRANSFER-LEARNING FROM TEXT

## ABSTRACT

Recent work has described neural-network-based agents that are trained to execute language-like commands in simulated worlds, as a step towards an intelligent agent or robot that can be instructed by human users. However, the instructions that such agents are trained to follow are typically generated from templates (by an environment simulator), and do not reflect the varied or ambiguous expressions used by real people. We address this issue by integrating language encoders that are pretrained on large text corpora into a situated, instruction-following agent. In a procedurally-randomized first-person 3D world, we first train agents to follow synthetic instructions requiring the identification, manipulation and relative positioning of visually-realistic object models. We then show how these abilities can transfer to a context where humans provide instructions in natural language, but only when agents are endowed with language encoding components that were pretrained on text-data. We explore techniques for integrating text-trained and environment-trained components into an agent, observing clear advantages for the fully-contextual phrase representations computed by the well-known BERT model, and additional gains by integrating a self-attention operation optimized to adapt BERT's representations for the agent's tasks and environment. These results bridge the gap between two successful strands of recent AI research: agent-centric behavior optimization and text-based representation learning.

## 1 INTRODUCTION

Developing machines that can follow natural human commands, particularly those pertaining to an environment shared by both machine and human, is a long-standing and elusive goal of AI (Winograd, 1972). Recent work has applied end-to-end, representation-learning-based methods to this challenge, where a neural-network-based agent is optimized to process language input, perceive its surroundings and execute appropriate movements jointly (Oh et al., 2017; Hermann et al., 2017; Chaplot et al., 2018). End-to-end learning promises a way to deal flexibly with the complexity of the physical, visual and linguistic world without relying on (potentially brittle) hand-crafted features, rules or policies. Nevertheless, the cost of this flexibility is the large number of environment interactions (samples) required for a randomly-initialized network to learn behaviour policies from raw experience. To make the approach feasible, many studies thus employ a synthetic language that is generated on demand from templates by the environment simulator (Chevalier-Boisvert et al., 2018; Jiang et al., 2019; Yu et al., 2018b;a). The studies that do combine both end-to-end learning with natural-language data do so in less realistic grid-like environments (Misra et al., 2017) or grant agents access to privileged global observations to make learning more tractable (Misra et al., 2018).

Here, we take a different learning-based approach to instruction-following that is robust to human commands. We train agent policies to respond to synthetic, template-based language but also endow them with powerful language encoders that are pretrained on natural language text. The synthetic instructions that our agents are trained to follow require mastery of 26 fine-grained motor actions in order to identify and manipulate visually-realistic models from the ShapeNet dataset (Chang et al., 2015) in a 3D room. The visual realism of the world makes it possible to elicit diverse natural human ways of instructing and/or referring to things, and to study the agents' robustness to this diversity. Unsurprisingly, we find that agents that are trained on template-based commands do not cope well with the diversity and variation of natural keyboard-typed language. In contrast, when powerful

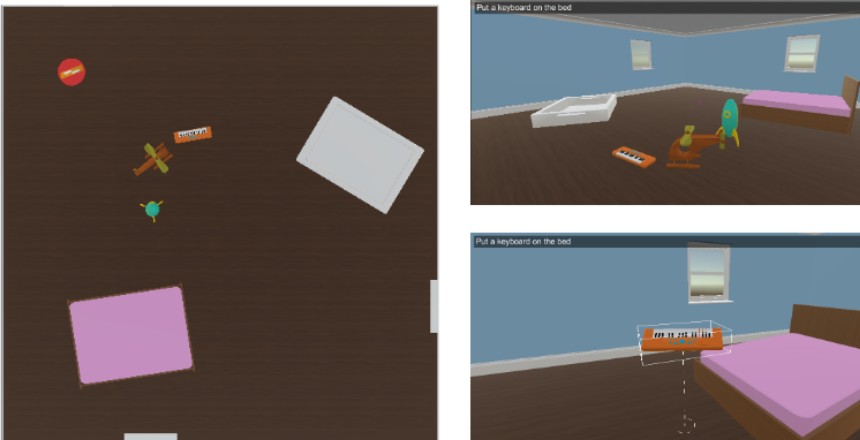

Figure 1: An episode of the putting task, seen from above (left) and from the agent's perspective (right). In each episode of training or evaluation, there are three randomly-coloured moveable objects, plus a pink bed and a white tray. Initial positions of all objects, doors, windows and the agent are random. To succeed in an evaluation trial, the agent must process an instruction given by a human (e.g. *"Put the musical keyboard on the bed"*, or *"Drop A Casio on the bed"*), identify the intended object of reference, move towards it and lift it, rotating if necessary, and then lower it into the specified location. Our best-performing agent succeeds at this evaluation with 72% accuracy.

| | Pixel observations | Memory and partial obs. | Object interaction | Object manipulation | Visual realism | Natural language |
|---|---|---|---|---|---|---|
| Oh et al. (2017) | ✓ | ✓ | | | | |
| Hermann et al. (2017) | ✓ | ✓ | | | | |
| Chaplot et al. (2018) | ✓ | ✓ | | | | |
| Misra et al. (2018) | ✓ | | ✓ | | ✓ | ✓ |
| Wu et al. (2018) | ✓ | ✓ | | | ✓ | |
| Jiang et al. (2019) | ✓ | | ✓ | ✓ | | |
| This work | ✓ | ✓ | ✓ | ✓ | ✓ | ✓ |

Table 1: Recent work on instruction-following in 3D environments (not exhaustive). By transfer-learning from simulation, our best-performing agents overcome unique combination of policy-optimization challenges (partial observability, memory, raw perception, object manipulation and relations) and can also interpret natural human commands.

pretrained language encoders are integrated into our agent, we find that agents can satisfy human instructions with substantially above-chance accuracy.

To better analyze the generalization that supports this robustness, we probe trained agents with specific modifications to their training instructions. We find that methods based on conventional (context-free) word embeddings support generalization that is driven by lexical similarity (executing `lift a vehicle` when trained to `lift a car`), but capture phrasal equivalence less well (failing at `put a plate on the container` when trained to `put the dish on the tray`). In contrast, methods that integrate contextual word representations support both types of generalization. We also find that robustness to synonyms can often be improved when pretrained encoders are combined with an additional (cross-modal) self-attention tuned to the agent's environmental objectives. Ablation experiments highlight the role that *WordPiece tokenization* (Schuster & Nakajima, 2012) plays in robustness to human instructions. This motivates the addition of *typo noise* to our training pipeline, which further improves the accuracy of the agent's responses.

Overall, our principal contributions are the following:

- We train an agent that can both interpret human language commands and overcome a similar range of behavioural and environmental challenges to state-of-the-art policy-learning approaches (Table 1).

- We develop the techniques of transfer-learning from a text representation-learning model to an embodied agent. Note that this is very different from transfer between language classification tasks (Collobert & Weston, 2008; Devlin et al., 2018); in our evaluations the agent must interpret unfamiliar instructions zero-shot (i.e. without additional learning steps), and must realize complex language-conditional behaviours (decoding $\approx 50$ appropriate actions) in spite of this uncertainty.

## 1.1 HIGH-LEVEL APPROACH

We experiment in a simulated room in the Unity[1] game engine (Fig 1). Importantly, the realism of the visual assets in this room enable human observers to recognize the objects (and hence refer to them in natural ways). Our experimental pipeline then involves the following steps:

1. Define a (language-dependent) task, including initial conditions for the simulated room, a template-based process for generating synthetic language instructions and a reward function for checking game states and exposing reward if the agent succeeds.
2. Train agents with different language encoders to solve the task when conditioned on the synthetic language produced by the environment generator.
3. Evaluate the trained agent on 1000 episodes of the task with the synthetic instructions replaced with synonyms or free-form commands typed by human annotators.

## 2 ARCHITECTURES FOR INSTRUCTION-FOLLOWING AGENTS

Since the object of our study is instruction-following, we consider an agent architecture that is as conventional as possible in aspects not related to language processing. It consists of standard components for computing visual representations of pixels, embedding text strings and predicting actions contingent on these and memory of past inputs.[2]

**Visual processing**   The visual observations received by the agent at each timestep are $96 \times 72 \times 3$ real-valued tensors, which are processed by a 3-layer residual convnet.

**Memory core**   The output of the visual processing is combined with language information according to a particular encoding strategy, as described below. In all conditions, some combination of vision and language input at each timestep passes into a LSTM memory core with hidden dim. 128.

**Action and value prediction**   The state of the memory core at each timestep is passed through a linear layer and softmax to compute a distribution over 26 actions. Independently, the memory state is passed to a linear layer to yield a scalar value prediction.

**Training algorithm**   The agent is trained using an importance-weighted actor-critic algorithm with a central learner and distributed actors (Espeholt et al., 2018).

## 2.1 LANGUAGE ENCODING

The agent must process both string observations in the simulated environment, during training, and human instructions, also encoded as strings, during evaluation. Its representations of language must be combined with visual information to make decisions about how to act. We compare various different ways of achieving this encoding. For methods that transfer knowledge from unsupervised text-based learning, we take weights from the well-known BERT model (Devlin et al., 2018), specifically the uncased BERT$_{\text{BASE}}$ model made available by the authors.[3]

**BERT + mean pool**   For a given input of $w$ words, BERT$_{\text{BASE}}$ returns $w$ context-dependent (sub)-word representations of 768 units. In this condition, a mean pooling operation is applied over the $w$ dimension to yield a single representation of dimension 768, which is concatenated with the flattened output of the visual processing module. This multi-modal representation is passed through a single layer MLP with $tanh$ activation and output dimension 128 before entering the memory core of the agent. We apply the standard BERT$_{\text{BASE}}$ WordPiece vocabulary (of size 30,000). Note

---

[1]Unity. http://unity3d.com.

[2]See Appendix C for details not included here.

[3]Available at `https://tfhub.dev/google/bert_uncased_L-12_H-768_A-12/1`

that WordPiece encodes language in terms of *subwords* (a mix of characters, common word chunks, morphemes and words) rather than the word-level vocabulary applied in more traditional neural language models.

**BERT + self-attention layer**  When applying BERT to text classification, performance can sometimes be improved by *fine-tuning* the weights in the BERT encoder to suit the task. Because of the large number of gradient updates required to learn a complex behaviour policy, fine-tuning the BERT weights in this way would cause substantial overfitting to the synthetic environment language. We therefore keep the BERT weights frozen, but experiment with an additional learned self-attention layer (Vaswani et al., 2017) to create a language encoder whose bottom layers are pretrained and whose final layer optimizes these representations to the present environment or tasks. This additional layer has 4 attention heads, and uses 64 dimensional key and value embeddings.

**BERT+ cross-modal self-attention**  We also consider a cross-modal self-attention layer of the type suggested by e.g. Tsai et al. (2019); Lu et al. (2019), which in our application provides an explicit pathway for the agent to bind visual experience to specific (contextual) word representations. In this case, we treat each of the output channels of the visual processing network as word-like entities, by passing them through a linear layer of output size 768 to match the BERT output. The embeddings for all words and the visual channels are then processed with a single self-attention layer, whose parameters are again learned with all other agent parameters.

**Pretrained (sub)-word embeddings**  Language classification experiments with BERT show the value of highly context-dependent (sub)-word representations, but transfer in that context is also possible with more conventional (context-independent) word embeddings (Collobert & Weston, 2008). To measure the effect of this distinction, we consider a simpler encoder based on the (context independent) input (sub-)word embeddings from BERT, which are also of dimension 768.[4] Taking a mean of these context-independent vectors would yield a word-order-invariant representation of language. We therefore process them with single-layer transformer with 4 attention heads. As in *BERT + mean pool*, this output is averaged, reduced by a single-layer MLP (from 768 to 128 units) and passed to the agent's core memory.

**Typo noise**  Finally, in one condition we introduce typo noise (each key character was replaced with that of an adjacent character on a standard keyboard, with a probability of 0.01) to the synthetic language strings produced by the environment, allowing the agent to learn to compensate during training. No modifications were made to the evaluation stimuli. Much previous work applies typing noise for language classifier robustness; see e.g. (Pruthi et al., 2019) for a recent survey.

We compare to various baselines designed to isolate specific components of these encoders.

**Random mean pool**  As a direct baseline for *BERT mean pool*, we consider an identical architecture but in which the contextual BERT embeddings are replaced by fixed random (subword-specific) vectors (also of dimension 768). The weights in the rest of the agent network are trained as in other conditions. We note that random sentence vectors are a competitive baseline on many language classification tasks (Wieting & Kiela, 2019).

**Word-level and WordPiece transformers**  In addition to pre-trained weights, a potentially important aspect of encoders based on BERT is WordPiece tokenization, which can afford greater ability to make sense of typos or rare words with familiar morphemes than in a more conventional word-level encoder. To isolate the effect of tokenization from that of pretrained weights, we compared two further encoders. In one, we split all input strings by whitespace and hash each word string to a unique index representing an input to a single-layer transformer with 4 attention heads and embedding size 768 (chosen to match BERT$_{\text{BASE}}$), the output of which is processed identically to the Random mean pool condition. We contrast this with an otherwise identical condition in which the WordPiece tokenization from BERT is applied rather than splitting on white space.

## 3  EXPERIMENTS

We experiment with a **lifting task**, which focuses on object identification, and a **putting task** which focuses on object relations and manipulation. In both tasks the locations of all objects and the

---

[4]These embeddings capture lexical similarity much like conventional word embeddings; a cosine metric produces a Spearman correlation of 0.49 with human ratings from the Simlex-999 dataset (Hill et al., 2015).

initial position and orientation of the agent are chosen at random on floating point scales, so it is highly unlikely that any two episodes are spatially identical (whether during training or evaluation). The action set consists of movements (`move-{forward,back,left,right}`), turns (`turn-{up,down,left,right}`), and object pick-up and manipulation with 4 DoF. The placement of objects is assisted by a visible vertical line. See Appendix C a full description.

## 3.1 LIFTING WITH HUMAN INSTRUCTIONS

In an episode of the lifting task, the environment selects two movable objects at random from a global object set, and generates a language instruction of the form `Lift a X`, where `X` is the correct name for (exactly) one of those objects. To achieve reward, the agent must locate the `X`, lift it more than 1m above the ground and keep it there for 5 time steps. If the reward function in the environment determines that these conditions are met, a positive reward of $+1$ is emitted and the episode ends. If a lifting takes place with the incorrect object, the episode also ends with reward 0. For a global set of recognisable objects, we use the *ShapeNet* dataset (Chang et al., 2015), which contains over 12,000 3D rendered models. For performance reasons we discard models with more than 8,000 vertices or an OBJ file size greater than 1 MB and consider only those tagged into a synset in the WordNet taxonomy. We use the name of the first lemma of that synset as a proxy for the name of the object. From these names, we selected 80 to for the environment, each referring to a set of object models with a minimum of 12 exemplars. See appendix D for more details.

For all language encoding strategies described above, the agent was able to learn the task, and a well-trained policy completed episodes in an average of $\approx 20$ timesteps with accuracy around 90%. The failure of agents to reach perfect performance on the training set is unimportant for the present study; we suspect that the agent's comparatively small convolutional network fails to perceive important distinctions between the more intricate ShapeNet models.

| Template lang. (training) | Synonym | Natural referring expression |
|---|---|---|
| Lift a flag | Lift a **banner** | Lift *the indian flag*
Lift *a flag*
Lift *the flag.* |
| Lift a pillow | Lift a **cushion** | Lift *a pillows*
Lift *a cushion*
Lift *a paper* |

Table 2: Example training (left) and test instructions in the lifting task.

| Template lang. (training) | D.O. synonym | I.O. synonym | D.O. & I.O. synonym | Natural instruction |
|---|---|---|---|---|
| Put a mug on the tray | Put a cup on the tray | Put a mug on the box | Put a cup on the box | *place mug in the basket*
*Keep the cup in atub*
*place the mug in a container*
*put the coffee mug in the box* |
| Put a train on the bed | Put a locomotive on the bed | Put a train on the bunk | Put a locomotive on the bunk | *Put the tractor on the bed*
*Move the train toy onto the bed*
*Place a toyvehicle on the bed*
*place the rail on tthe bed* |

Table 3: Example training (left) and test instructions in the putting task. Underlined words are examples of synonyms, italics indicate entire phrases provided by human annotators.

We consider two evaluation settings. In the **synonym** evaluation, we ran the environment for 1,000 episodes with the noun in the environment template instruction replaced by a synonym (`Lift a` $X$ becomes `Lift a` $X^*$ where $X \approx X^*$). The synonyms were provided by native English speaking subjects. In the **natural referring expression** evaluation, we gave 40 annotators access to a room containing a single ShapeNet model via a crowd-sourcing platform. We asked them to write down what they found in the room and then hit a button that restarted the environment with a new object in the room.[5] As illustrated in Table 2, unlike the synonyms, the natural referring expressions involve variation in articles as well as nouns (*a pencil* might become *the pen*), may include spelling mistakes or typos, can refer entirely incorrectly to the intended object (if the subject fails to recognize the ShapeNet model), but may also match the training instruction exactly. Moreover, unlike the synonym test, there are $30 - 40$ natural referring expressions for each of the 80 environment nouns, from which we sample randomly when evaluating the agent (again on 1,000 evaluation espisodes).

---

[5]See Appendix B for the full list of synonyms and annotator instructions.

## 3.2 PUTTING OBJECTS ON OTHER OBJECTS WITH HUMAN INSTRUCTIONS

The strength of models like BERT is their ability to combine lexical representations into phrasal or sentence representations. To study this capacity in the context of instruction-following, we devised a **putting task** involving the verb 'to put', which, in the imperative (put the cup on the tray) takes two arguments, the *direct object (D.O.)* cup and the *indirect object (I.O.)* tray. In terms of the behaviours required, the putting task focuses on manipulation and object relations rather than object identification or reference. The environment was configured to begin each episode with three randomly-chosen moveable objects and two larger immovable objects (a bed and a tray), each randomly positioned in the room. In each episode of this task, the agent receives an instruction Put a D.O. on the I.O., where D.O. is any of the three moveable objects (chosen from a global set of ten) and I.O. is either bed or tray. The environment checks whether an instance of D.O. is at rest (and not held by the agent) on top of the I.O., returning a positive reward $+1$ if so and ending the episode. If the object D.O. is placed on something other than a I.O., or if another movable object is placed on the bed or the tray then again the episode ends immediately with reward $0$.[6]

As before, we first trained all agents on the putting task with synthetic environment language instructions. Training a policy on this task with reinforcement learning required a bespoke task curriculum (see Appendix C for details); a well-trained policy completes each episode in an average of $\approx 50$ actions/timesteps. To gather the evaluation stimuli, we again crowd-sourced humans to provide both natural synonyms for each of the 12 objects in the global set for this task and, in this case, entirely free-form natural human instructions. To obtain natural instructions, we instantiated an environment with only one of the global set of moveable objects, coloured red, and one of either the bed or the tray, coloured white, and asked subjects to *ask somebody to place the red object on top of the white object without mentioning their color.* From these instructions, we defined four evaluations, illustrated in Table 3: **D.O. synonym**, **I.O. synonym** and **D.O. & I.O. synonym**, in which particular parts of the original template command were replaced with synonyms, and **Natural instruction**, the fully free-form human instruction, which can include orthographic errors and misidentified objects.

## 3.3 DISCUSSION OF RESULTS

| Model | Template language (training) | Synonym | Natural referring expression |
|---|:---:|:---:|:---:|
| Random 'lifting' act | 0.5 | 0.50 | 0.50 |
| Random-embedding + MP | 0.86 | 0.49 | 0.48 |
| Word-level Transf. | 0.91 | 0.58 | 0.61 |
| WordPiece Transf. | 0.89 | 0.57 | 0.66 |
| Word-level Transf. + TN | 0.86 | 0.62 | 0.70 |
| WordPiece Transf. + TN | 0.86 | 0.64 | 0.69 |
| Word embeddings + Transf. | 0.89 | 0.62 | 0.64 |
| BERT + MP | 0.93 | **0.77** | **0.76** |
| BERT + SA | 0.91 | 0.70 | 0.70 |
| BERT + CMSA | 0.88 | 0.68 | 0.67 |
| BERT + CMSA + TN | 0.89 | 0.74 | 0.73 |
| Multitask BERT + CMSA + TN | 0.87 | 0.74 | 0.73 |

Table 4: Accuracy of different models on training instructions, the synonym evaluation and the natural referring expression evaluation. MP: mean pool, SA: self-attention layer, CMSA: cross-modal self-attention layer, TN: typo-noise. Multitask: single agent trained on both 'lifting' and 'putting' tasks. Scores show mean across 1,000 episodes (with instructions randomly chosen by the environment generator).

The accuracies for both lifting tasks are presented in Table 4 and for putting tasks in Table 5. The results reveal the following main effects of language encoding on model performance:

**Substantial transfer from text requires contextual encoders** Agents with weights that are pre-trained on text data exhibit substantially higher accuracy on both the lifting and the putting tasks. This effect is greatest in the more focused synonym evaluations, but also holds for the the free-form human instructions. A small transfer effect can be seen by comparing the *word embeddings + Transformer condition* (62% accuracy on the synonym evaluation, lifting task and 57% accuracy

---

[6]We found that ending the episode in such cases made learning much faster.

| Model | Template lang (training) | D.O. synonym | I.O. synonym | D.O. & I.O. synonym | Natural instruction |
|---|---|---|---|---|---|
| Random 'putting' act | 0.17 | 0.17 | 0.17 | 0.17 | 0.17 |
| Random-embedding + MP | 0.99 | 0.59 | 0.22 | 0.15 | 0.39 |
| Word-level Transf. | 0.99 | 0.48 | 0.09 | 0.03 | 0.36 |
| WordPiece Transf. | 0.98 | 0.30 | 0.24 | 0.10 | 0.42 |
| Word-level Transf. + TN | 0.97 | 0.61 | 0.27 | 0.17 | 0.45 |
| WordPiece Transf. + TN | 0.97 | 0.36 | 0.36 | 0.13 | 0.56 |
| Word embeddings + Transf. | 0.99 | 0.57 | 0.47 | 0.36 | 0.55 |
| BERT + MP | 0.99 | 0.94 | 0.74 | 0.57 | 0.49 |
| BERT + SA | 0.99 | **0.96** | 0.69 | 0.56 | 0.54 |
| BERT + CMSA | 0.99 | 0.93 | 0.67 | 0.47 | 0.43 |
| BERT + CMSA + TN | 0.98 | 0.88 | **0.79** | **0.70** | **0.70** |
| Multitask BERT + CMSA + TN | 0.98 | **0.96** | 0.75 | 0.68 | 0.66 |

Table 5: Accuracy of different models on the putting task when different parts of instructions of the form `"Put a [D.O.] on the [I.O.]"` are replaced with synonyms. Underlined words in model names indicate pre-trained weights from text-based training. MP: mean pool, SA: self-attention layer, CMSA: cross-modal self-attention layer, TN: typo-noise. Multitask: single agent trained on both 'lifting' and 'putting' tasks. The evaluation task covers episodes in which `[I.O.]` is either 'bed' or 'tray'. Scores show mean across 1,000 random episodes.

on the D.O. synonym evaluation, putting task) with the *WordPiece Transformer* (57% and 20%). However, overall the transfer effect is much stronger in the case of the full context-dependent BERT representations. On the same two evaluations, *BERT + mean pool* achieves 77% and 94% accuracy respectively. The gains from transferring via BERT representations vs. just (sub)word embeddings are greatest for the (longer) putting instructions than for the lifting instructions, and greatest of all in the *D.O & I.O. synonym evaluation*. These are cases where one would expect the marginal value of powerful sentential (rather than just lexical) representations should be greatest.

**Tuning via self-attention layers (with typo noise) helps** Interestingly, we find that tuned self-attention layers do not improve generalization performance over using BERT and mean pooling. This may be simply because the additional layers cause a degree of overfitting to the template environment language during training. However, typo noise mitigates this issue, so that the strongest evaluation performance on the putting tasks overall is observed with a combination of a tuned cross-modal self-attention layer and typo-noise training.[7] Indeed, the value of typo noise as a regularizer can be seen by the fact that it improves the robustness of agents with tuned self-attention layers even in the *synonym* evaluations (for both lifting and

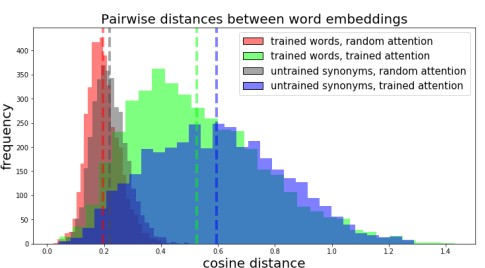

Figure 2: Self-attention layers learn to pull-apart both training nouns *and their synonyms* as the agent learns the putting task with synthetic environment language.

putting), which do not involve any typos. Thus, the *BERT + CMSA + TN* model performs better than all others on two of the three synonym evaluations in the putting task.

One way in which the appropriately-tuned self-attention layer might make the agent more robust to synonyms in this case is by spreading out task-relevant object-nouns in the agent's language representation space (leaving those words closer to synonyms than to potential confounding words). The degree to which this happens when the object-nouns and their synonymns in our environments are passed first through BERT and then through a (BERT + SA) agent-tuned self-attention layer (compared to passing through the same layer but with random weights) is shown in Fig 2.

**WordPiece tokenization adds robustness** In the two evaluations involving natural language instructions from humans, a comparison of *Word-level Transformer* and *WordPiece Transformer* shows that some robustness is obtained simply from WordPiece encoding, which in turn must play

---

[7]See Appendix A to compare BERT-based architectures with and without typo-noise.

some part in all BERT-based conditions (e.g. improving from 61% to 66% in the natural refer-ring expression evaluation and from 36% to 42% in the natural instruction putting evaluation). As mentioned above, BERT-based agents with WordPiece encoding are particularly robust to human instructions when trained with typo noise, and this is most effective when combined with tuned self-attention layers. This makes intuitive sense, as a learnable self-attention layer should provide the agent with more flexibility to learn to correct for typos during training. Indeed, on the natural instruction evaluation of the putting task, where typos or spelling errors are most common, the cross-modal attention agent trained with typo noise achieves 70% accuracy. Note that 100% accuracy on this evaluation may be impossible, even for humans, because visual ambiguity or human error mean that instructions can sometimes refer in entirely mistaken ways to the objects in the room.

**Multitask learning is possible** The performance of an agent trained on both the putting and lifting tasks is not substantially lower than agents that are specialized to each of the tasks individually.

## 4 RELATED WORK

Most closely related to our work is an experiment by Chan et al. (2019), who showed how an agent trained with *InferLite* sentence representations (Kiros & Chan, 2018) can be robust to synonym re-placements in template instructions. The task itself involves object identification in the VizDoom environment (Kempka et al., 2016), which requires only 3 motor actions. Our work develops this insight substantially, applying a similar approach to a visually-realistic environment requiring fine-grained object manipulation, integrating context-dependent pre-trained models with subword tok-enization (BERT), analysing architectures and training strategies for integrating such models and extending from synonym replacements to free-form instructions typed by humans.

Much recent work applies deep learning and policy optimization in end-to-end approaches to learn-ing instruction following (Chaplot et al., 2018; Oh et al., 2017; Bahdanau et al., 2018; Chevalier-Boisvert et al., 2018; Yu et al., 2018b;a; Jiang et al., 2019). As noted in the introduction, these studies do not typically involve natural language. Our work differs from studies that do (Misra et al., 2017; 2018) in several ways. First, we consider generalization and robustness (rather than performance when trained on a set of natural language commands). Second, we apply our method to learn much more complex policies requiring object manipulation and positioning and mastery of fine-grained action sets. Our approach makes learning such policies possible because we combine (potentially infinite) template-based RL training with text-based pretraining and transfer.

A limitation of all studies above is the reliance on simulation. Both object identification and manip-ulation are likely far harder in reality, and it remains to be seen whether our methods scale to robot language understanding (Tellex et al., 2011; 2012; Walter et al., 2014). See also (Anderson et al., 2018; Wang et al., 2019) for recent improvements to visual realism in simulated environments.

Finally, there is a long history of building in knowledge about the structure of language and/or its environment into instruction-following systems rather than learning it end-to-end. In Winograd (1972)'s SHRDLU, syntactic modules parsed the language input into a logical form, and hand-written rules were applied to connect such forms to the environment. More recent pipeline-based approaches use learning algorithms to map language to a program that can then interface with a planner (Chen & Mooney, 2011; Matuszek et al., 2013; Wang et al., 2016), and/or a controller, both of which may have priviledged information about how the world connects to the program. It is likely that pretrained language encoders could add robustness to parts of these approaches, much as they do here. Our focus on end-to-end learning, however, is motivated by the intuition that it may eventually scale or adapt more flexibly to arbitrary environments or problems than pipeline approaches.

## 5 CONCLUSIONS

In this work, we have developed an agent that can follow natural human instructions requiring the identification, manipulation and positioning of visually-realistic assets. Our method relies on zero-shot transfer from template language instructions to those given by human annotators when asked to refer and instruct in natural ways. The results show that, with powerful pretrained language en-coders, this transfer effect is sufficiently strong to permit decoding of complex language-dependent motor behaviours, despite the shift in distribution of the agent's input. More generally, we hope that this contribution serves to bring research on text-based and situated language learning closer

together. To facilitate research in this direction, we provide our dataset of natural instructions and referring expressions aligned to ShapeNet models.

We have explored only a tiny fraction of the many possible ways to transfer knowledge from a text corpus to a situated environment. For instance, our approach involves freezing rather than fine-tuning the BERT encoder weights to our desired behaviour policy, to avoid overfitting, but techniques such as knowledge distillation (Hinton et al., 2015) could point to more elegant ways to learn jointly from text and environmental experiences. Moreover, we have focused on BERT, but improvements may be possible by applying alternative general-purpose language encoders, such as GPT-2 (Radford et al., 2019), Roberta (Liu et al., 2019) and Transformer XL (Dai et al., 2019).

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

## A  FULL EXPERIMENTAL RESULTS

The results presented in charts here are those shown in the tables in the paper, with two additional conditions involving typo noise.

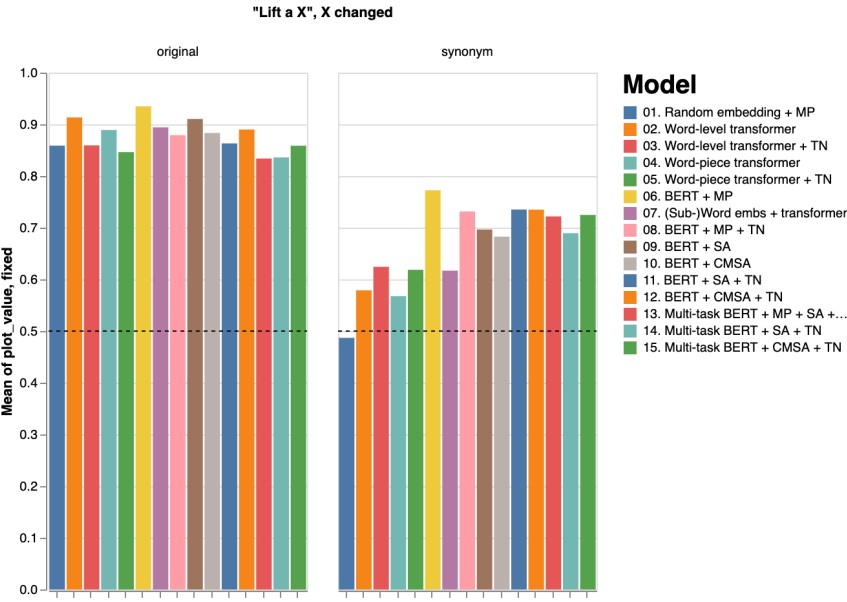

Figure 3: **Lifting task, synonym evaluation**. Left: accuracy of different agents over 1,000 episodes when no change is made to the environment template instruction. Right: accuracy of different agents when a synonym is introduced. Dotted line indicates score of an agent that carries out the correct behaviour but with random objects from the room.

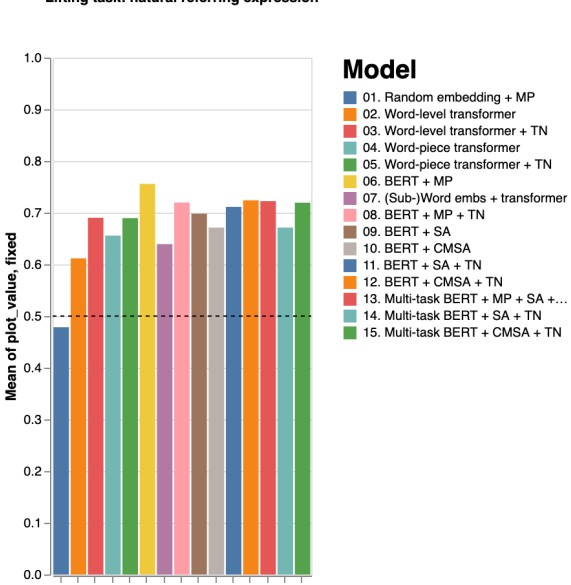

Figure 4: **Lifting task, natural referring expression evaluation**. Performance of agents on 1,000 evaluation episodes when part of the environment template lifting instruction is replaced by a natural referring expression from human annotators. Dotted line indicates score of an agent that carries out the correct behaviour but with random objects from the room.

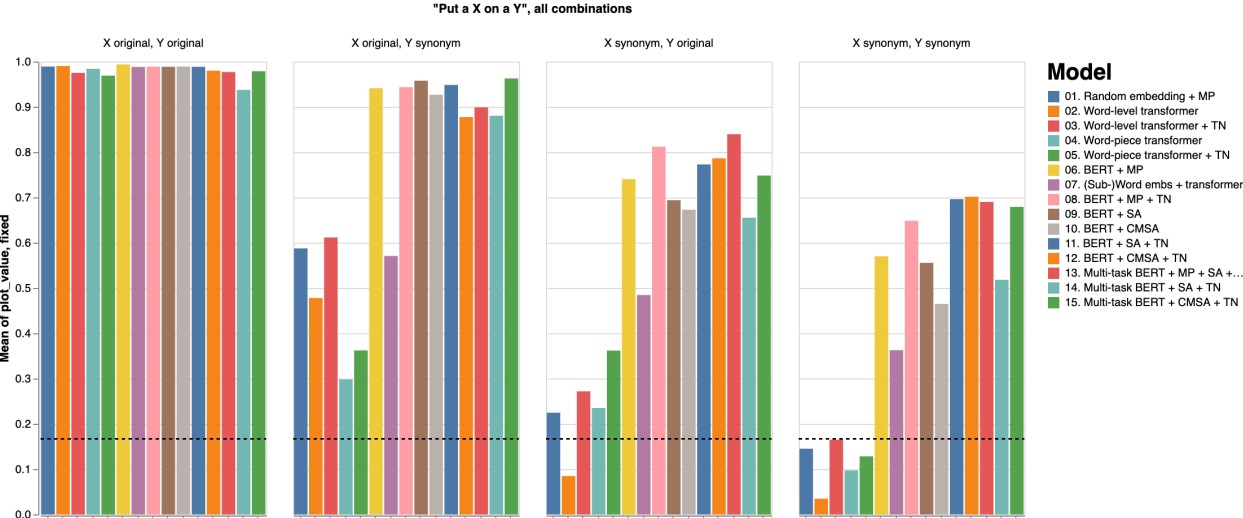

Figure 5: **Putting task, synonym evaluation**. Performance of agents on 1,000 evaluation episodes when (left) no change is made to the template environment instruction and (rightmost three) different parts of the template environment instruction are replaced with synonyms. Dotted line indicates score of an agent that carries out the correct behaviour but with random objects from the room.

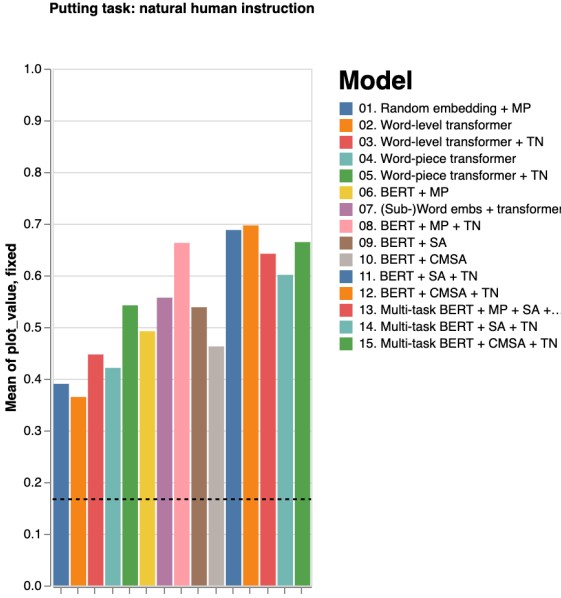

Figure 6: **Putting task, natural instruction evaluation**. Performance of agents on 1,000 evaluation episodes when the full environment template putting instruction is replaced by a natural instruction from human annotators. Dotted line indicates score of an agent that carries out the correct behaviour but with random objects from the room.

# B    INSTRUCTIONS TO HUMAN ANNOTATORS

*Human annotators use a keyboard and mouse to control a player in the environment simulator, as is standard in first-person video games. The annotators were given the following instructions as part of the each annotation task.*

## B.1    NATURAL REFERRING EXPRESSIONS

This is a task called **Name The Object**. You will find yourself in a room containing a single object. Please move around the room to get a good view of the object. When you know what the object is:

1. Hit Enter
2. Type the name of the object
3. Hit Enter again

Examples of good responses:

1. *A kettle*
2. *Some trousers*
3. *A pair of scissors*
4. *A tennis ball*

Please don't describe the object. Just write down what you see, with an article like 'a' or 'some' if appropriate.

Example of bad responses

1. *A brown ball*
2. *A large piano with long black legs*
3. *A small thing with lumps on the side*

You should not need more than 4-5 different words, and most objects will require just 1 or 2 words to name.

Sometimes, you might be unsure what the object is. If that's the case, just make your best guess.

## B.2    FULL HUMAN INSTRUCTIONS

This is a task called **Ask to put**. You will find yourself in a room. Your job is to imagine giving an instruction to somebody else so that that person puts the red object on to the white object. Move around the room to get a good look at what the two objects are. When you are ready to give your instruction:

1. Hit Enter
2. Type your instruction
3. Hit Enter again

Examples of good instructions:

1. *Place the cup onto the table*
2. *Put the ball on the plate*
3. *Move the pencil onto the box*

**Words to avoid** Your instruction must not contain words for colours or other properties of the object. Please do not use the words red, white, scarlet, dark, large etc. in your instruction. Instead, refer to objects by their name as you recognise them. If you don't recognise what an object is, just make your best guess.

Examples of bad instructions:

1. *Put the red thing on the table*
2. *Put the large object on the small object*
3. *Put the small round thing on the chest*

Keep your language varied. Try to use various ways to express your instruction in different episodes to keep things interesting.

## C  AGENT ARCHITECTURE AND TRAINING DETAILS

To process visual input, the agent uses a residual convolutional network with $64, 64, 32$ channels in the first, second and third layers respectively and 2 residual blocks in each layer.

As the agent learns, actors carry replicas of the latest learner network, interact with the environment independently and send trajectories (observations and agent actions) back to a central learner. The learning algorithm (Espeholt et al., 2018) modifies the learner weights to optimize an actor-critic objective, with updates importance-weighted to correct for differences between the actor policy and the current state of the learner policy. We used a learning rate of $0.0005$, a learner batch size of $32$, an agent unroll length of $50$, a discounting factor of $0.99$, an epsilon (epsilon-greedy policy) of $1e^{-6}$ and an *entropy cost* of $0.0003$ (Mnih et al., 2016). We use an Adam optimizer (Kingma & Ba, 2014) with $\beta_1 = 0.90$ and $\beta_2 = 0.95$.

In order to train the agent on the *putting task*, it was necessary to combine episodes of the task itself with episodes of simpler tasks, in a form of curriculum (although learning in parallel on all tasks). In particular, we trained it concurrently on a *lifting task* that involved the same moveable objects as the putting task (so that the agent could receive signal about their names without relying on a complete act of putting). We also found it beneficial to add a *put-near* task to the curriculum, where the instructions were of the form `Put a cup near the tray` rather than `Put a cup on the tray`, and a reward was emitted if the agent moved the cup a short distance from the tray and placed it on the ground.

In the *lifting task*, training the agent requires approximately 200 million frames of experience (approximately 7 million episodes), which takes about 24 hours with 250 actors on GPU. In the *putting task*, training in each condition was stopped after 30 million episodes, approximately three days of training.

| Body movement actions | Movement and grip actions | Object manipulation |
|---|---|---|
| NOOP | GRAB | GRAB + SPIN_OBJECT_RIGHT |
| MOVE_FORWARD | GRAB + MOVE_FORWARD | GRAB + SPIN_OBJECT_LEFT |
| MOVE_BACKWARD | GRAB + MOVE_BACKWARD | GRAB + SPIN_OBJECT_UP |
| MOVE_RIGHT | GRAB + MOVE_RIGHT | GRAB + SPIN_OBJECT_DOWN |
| MOVE_LEFT | GRAB + MOVE_LEFT | GRAB + SPIN_OBJECT_FORWARD |
| LOOK_RIGHT | GRAB + LOOK_RIGHT | GRAB + SPIN_OBJECT_BACKWARD |
| LOOK_LEFT | GRAB + LOOK_LEFT | GRAB + PUSH_OBJECT_AWAY |
| LOOK_UP | GRAB + LOOK_UP | GRAB + PULL_OBJECT_CLOSE |
| LOOK_DOWN | GRAB + LOOK_DOWN | |

## D  FURTHER ENVIRONMENT DETAILS

For all experiments, the environment is a Unity room of dimension 4m x 4m. The walls, floor and ceiling are always the same color, but we add a window and door (positioned randomly per episode) to give some sense of absolute location to the agent.

When importing ShapeNet models, we use the scaling provided in the metadata, unless it it is very small (less than 0.000001), in which case we interpret the coordinates in the OBJ file as meters. All objects have rigid bodies with collision meshes generated using Unity's built in MeshCollider, with convex set to true. The masses of all movable objects are set to 1 kg, so that our avatar has enough strength to pick all of them up (except for beds and trays, which are made kinetic).

When selecting ShapeNet models, for performance reasons we discarded all models with a vertex count higher than 8000, and an OBJ file size greater than 1 MB. The native ShapeNet category names are often not natural everyday names (for instance, they can be highly specific, like "dual shock analog controller"). To mitigate this, we used ShapeNet's WordNet tags, grouping models into categories according to WordNet synsets and assigning the name of the first synset lemma to the category.

Because depth-perception is challenging without binocular vision, the agent is assisted in manipulation by a visual guide (bottom-right) that highlights objects within grasping range and drops a vertical line from held objects.

Full lists of objects and synonyms are on the following pages.

| name | synonym | unique models | example shapenet model id | wordnet synset |
|---|---|---|---|---|
| chest of drawers | cupboard | 241 | 793aa6d322f1e31d9c75eb4326997fae | n3018908 |
| table | desk | 199 | db80fbb9728e5df343f47bfd2fc426f7 | n4386330 |
| chair | seat | 167 | 6bcabd23a81dd790e386ecf78eadd61c | n3005231 |
| sofa | couch | 158 | 11d5e99e8faa10ff3564590844406360 | n4263630 |
| television receiver | aerial | 147 | cebb35bd663c82d3554a13580615ae1 | n4413042 |
| lamp | light | 138 | 62ab9c2f7b826fbcb910025244eec99a | n3641940 |
| desk | bench | 98 | a2c81b5c0ce9231245bc5a3234295587 | n3184367 |
| box | tin | 81 | 15b5c54a8ddc84353a5acb1c5cbd19ff | n2886585 |
| vase | flute | 78 | 415a96bfbb45b3ee35836c728d324152 | n4529463 |
| bed | bunk | 73 | bbd23cc1e56a24204e649bb31ff339dd | n2821967 |
| floor lamp | lamp | 73 | 54017ba00148a652846fa729d90e125 | n3371905 |
| cabinet | cupboard | 68 | f5024636f3514eb51d0662c550f8f994 | n2936496 |
| book | magazine | 67 | c8691c86e110318ef2bc9da1ba799c60 | n6422547 |
| pencil | pen | 67 | 839923a17af633dfb29a5ca18b250f3b | n3914323 |
| laptop | computer | 66 | 20d42e934260b59c53c0c910fd6231ef | n3648120 |
| monitor | screen | 58 | 4d11b3c781f36fb3675041302508f0e1 | n3787723 |
| coffee table | table | 53 | e21cadab660eb1c5c71d82cf1efbe60a | n3067971 |
| picture | image | 52 | 2c31ea08641b076d6ab1a912a88dca35 | n3937282 |
| bench | chair | 52 | c6706e8a2b4efef5133db57650fae9d9 | n2832068 |
| painting | portrait | 51 | 6e51501eee677f7f73b3b0e3e8724599 | n3882197 |
| rug | carpet | 44 | b49fda0f6fc828e8c2b8c5618e94c762 | n4125115 |
| shelf | surface | 43 | 482f328d1a777c4bd810b14a81e12eca | n4197095 |
| switch | flower | 42 | 7f20daa4952b5f5a61e2803817d42718 | n4379457 |
| plant | shrub | 42 | 94a36139281d1837465e08d496c0420f | n17402 |
| stool | seat | 41 | 1b0626e5a8bf92b3945a77b945b7b70f | n4334034 |
| bottle | jug | 40 | a429f8eb0c3e6a1e6ea2d79f658bbae7 | n2879899 |
| loudspeaker | speaker | 37 | 82b9111b3232904eec3b2e05ce8fd39b | n3696785 |
| electric refrigerator | fridge | 36 | 392f9b7a08c8ba8718c6f74ea0d202aa | n3278824 |
| toilet | urinal | 34 | 5d567a0b5b57d8ab8b6558e44187a06e | n4453655 |
| dining table | table | 29 | 82a1545cc0b3227ede650492e45fb14f | n3205892 |
| poster | picture | 28 | de74ab90cc9f46af9703672f184b66db | n6806283 |
| wall clock | clock | 28 | 1e2ea05e566e315c35836c728d324152 | n4555566 |
| cellular telephone | mobile | 26 | 85a94f368a791343985b19765176f4ab | n2995984 |
| person | human | 25 | 2c3dbe3bd247b1ddb19d42104c111188 | n5224944 |
| cup | mug | 24 | 542235fc88d22e1e3406473757712946 | n3152175 |
| stapler | clipper | 24 | 376eb047b40ef4f6a480e3d8fdbd4a92 | n4310635 |
| mirror | reflector | 24 | 36d2cb436ef75bc7fae7b9efb5c3bbd1 | n3778568 |
| toilet tissue | toilet paper | 24 | 6658857ea89df65ea35a7666f0cfa5bb | n15099708 |
| desktop computer | pc | 24 | 102a6b7809f4e51813842bc8ef6fe18 | n3184677 |
| table lamp | desk light | 23 | 85b52753cc7e7207cf004563556ddb36 | n4387620 |
| flag | banner | 22 | ac50bd1ed53b7cb9cec8a10ad1c084eb | n3359749 |
| armchair | settee | 21 | 806bce1a95268006ecd7cae46ee113ea | n2741540 |
| cupboard | wardrobe | 21 | 3bb80aa0267a12bed00bff798ed59ff5 | n3152990 |
| pen | pencil | 20 | 628e5c83b1762320873ca101f05858b9 | n3913116 |
| bag | sack | 20 | b2c9ac70c58c90fa6dd2d391b72f2211 | n2776843 |
| soda can | drink | 20 | 16526d147e837c386829bf9ee210f5e7 | n4262696 |
| sword | knife | 20 | 555c17f73cd6d530603c267665ac68a6 | n4380981 |
| bookshelf | shelf | 20 | 586356e8809b6a678d44b95ca8abc7b2 | n2874800 |
| fireplace | burner | 19 | df1bd1065e7f7cde5e29ce2c9d37b952 | n3351301 |
| curtain | drape | 19 | 6f3da555075ec7f9e17e45953c2e0371 | n3155743 |
| battery | cell | 18 | 62733b55e76a3b718c9d9ab13336021b | n2813606 |
| bookcase | bookshelf | 18 | bc80335bbfda741df1783a44a88d6274 | n2874241 |
| pizza | pie | 18 | caca4c8d409cddc66b04c0f74e5b376e | n7889783 |
| hammer | mallet | 17 | a49a6c15dcef467bc84c00e08b1e25d7 | n3486255 |
| microwave | oven | 16 | b3bf04a02a596b139220647403cfb896 | n3766619 |
| cereal box | oatmeal | 16 | dc394c7fdea7e07887e775fad2c0bf27 | n3001610 |
| food | meal | 16 | d92f30d9e38cf61ce69bbdea737daae6 | n21445 |
| screen | monitor | 16 | 4d7fb62f0ed368a18f62bdf4e9082924 | n4159912 |
| glass | beaker | 16 | 89cf9af7513ecd0947bdf66811027e14 | n3443167 |
| wine bottle | magnum | 15 | e101cc44ead036294bc79c881a0e818b | n4599016 |
| lamppost | streetlight | 15 | 9cf4a30ab7e41c85c671a0255ba06fe5 | n3642472 |
| oven | cooker | 15 | 69c57dd0f6abdab7ac51268fdb437a9e | n3868196 |
| camera | polaroid | 14 | 48e26e789524c158e1e4b7162e96446c | n2946154 |
| globe | world | 14 | 3e97d91fda31a1c56ab57877a8c22e14 | n3445436 |
| cd player | stereo | 14 | fe90d87deaa5a8a5336d4ad4cfab5bfe | n2991759 |
| wardrobe | cupboard | 14 | cb48ec828b688a78d747fd5e045d7490 | n4557470 |
| bible | text | 14 | 6ad85ddb5a110664e632c30f54a9aa37 | n6434286 |
| pillow | cushion | 13 | 8b0c10a775c4c4edc1ebca21882cca5d | n3944520 |
| mug | cup | 13 | ea33ad442b032208d778b73d04298f62 | n3802912 |
| chandelier | light | 13 | 37d81dd3c640a6e2b3087a7528d1dd6a | n3008889 |
| chessboard | checkers | 13 | ec6e09bca187c688a4166ee2938aa8ff | n3017971 |
| calculator | computer | 13 | 387e59aec6f5fdc04b836f408176a54c | n2942270 |
| dishwasher | washing machine | 12 | ff421b871c104dabf37a318b55c6a3c | n3212662 |
| mattress | mat | 12 | f30cb7b1b9a4184eb32a756399014da6 | n3736655 |
| basket | bag | 12 | 91b15dd98a6320afc26651d9d35b77ca | n2805104 |
| pencil sharpener | sharpener | 12 | 287b8f5f679b8e8ecf01bc59d215f0 | n3914833 |
| candle | lantern | 12 | cf1b637d9c8c30da1c637e821f12a67 | n2951508 |
| bowl | dish | 12 | 594b22f21daf33ce6aea2f18ee404fd5 | n2884435 |
| clock | watch | 12 | 299832be465f4037485059ffe7a2f9c7 | n3050242 |
| coat hanger | hanger | 12 | 5aa1d74d04065fd98a7103092f8e8a33 | n3061905 |

Table 6: The 80 ShapeNet category names and their sizes, corresponding WordNet synsets and an example model id for a category exemplar used in the *lifting task*. In the extra materials we provide a full list of ShapeNet model ids in each category.

|  | **Environment word** | **Synonym** |
|---|---|---|
| Immovable objects | tray | box |
|  | bed | mattress |
| Movable objects | boat | ship |
|  | hairdryer | dryer |
|  | racket | bat |
|  | bus | coach |
|  | rocket | spaceship |
|  | car | automobile |
|  | plane | aeroplane |
|  | mug | cup |
|  | robot | android |
|  | train | locomotive |
|  | keyboard | piano |
|  | helicopter | airplane |
|  | candle | lamp |

Table 7: The two immovable and ten movable objects in the putting experiment. To begin each episode, both immovable objects and three randomly-selected movable objects are randomly positioned in the room.

