# OpenReview forum: "Robust Instruction-Following in a Situated Agent via Transfer-Learning from Text"
_ICLR.cc/2020/Conference — Reject_

### Official Review · AnonReviewer3 · 2019-10-22
**Official Blind Review #3**

**Rating:** 6

**Review:**

This work proposes applying natural language encoders pre-trained on a large text corpora (e.g. BERT) to training agents to follow natural language instructions in a simulated environment. Overall, I greatly enjoyed reading this paper: clear exposition of the idea, sensible model architecture, reasonable baselines and good experimental performance. I only have minor questions & feedback, see below.

Pros
- Well designed experiments with sensible baselines.
- Strong transfer learning results.
- Illuminating analysis where using BERT indeed performs better on capturing phrasal and sentence-level equivalence in natural language instructions.

Questions
- In Table 5, BERT doesn't give performance improvement on natural instruction (over Word embedding+Transformers) until BERT+CMSA+TN. Why do you think this is the case? To phrase this in a different way, why do you think BERT+MP doesn't perform well on this?
- Are the results in Figure 2 computed from a BERT+MP model or a BERT+CMSA model?
- In the lifting results in Table 4, why doesn't BERT+CMSA+MP outperform BERT+MP?

------

Updates:

Having read other reviewers' comments and also the authors' response, I would also like to call into question the difficulty of the experiments in the paper -- for the lifting task, the model is always presented with a command "Lift X", and essentially only needs to identify the correct object out of two at test time. Also, for the putting task, the model only needs to disambiguate between 6 possible combinations (3 movable and 2 fixed objects). Especially with a sophisticated model like BERT, I would have liked to see tasks where human instructions are more complex than this simple task. Hence, I'm changing my score to 6.

**Experience Assessment:**

I have published one or two papers in this area.

**Review Assessment: Checking Correctness Of Derivations And Theory:**

I assessed the sensibility of the derivations and theory.

**Review Assessment: Checking Correctness Of Experiments:**

I assessed the sensibility of the experiments.

**Review Assessment: Thoroughness In Paper Reading:**

I read the paper at least twice and used my best judgement in assessing the paper.

---

### Official Review · AnonReviewer1 · 2019-10-24
**Official Blind Review #1**

**Rating:** 3

**Review:**

The authors present a method of transferring template-based instruction following agents to natural language instructions by using language encoders trained on large text corpora. They explore different ways of combining text-based language encoders with visual representations and compare them. They find that contextual phrase-based representations learned by BERT significantly improve the performance on natural language instructions.

Strengths:
- The paper is written well, it is easy to understand and follow.
- The task setup is good, authors collect natural language instruction data from humans.
- The paper presents several language encoding methods for the task and systematically evaluates them in a scientific manner.
- The experimental results indicate that it is possible to transfer an agent trained on template-based instructions to natural language instructions using language models trained on large text corpora. It is not necessary to train the agent on natural language instructions. I find this result important and useful.

Weaknesses:
- The paper lacks significant technical novelty. It essentially combines known reinforcement learning based instruction following agents with known language models. The different ways of combining language encoding with visual representations are either trivial or adapted from prior work.
- A major concern is that the natural language instructions considered in the paper do not have much diversity with respect to language. The paper only considers lifting and putting tasks and trains a separate model for both the tasks.
-- The lifting task always uses the verb 'lift' and replaces the object word with synonyms or referring expressions. There are 80 objects in the lifting task and I suspect there are very few referring expressions for these objects and they mostly involve a synonym. Furthermore, at test time, the agent only needs to distinguish between 2 objects. The performance with random embedding is around 50% for this task and the best model is around 76% which means the agent is not recognizing the correct object around 50% of the time.
-- For the putting task, authors consider synonyms for object words and natural instructions which involve changing the verb ‘put’. It seems like humans mostly use only 4 verb words for this task, ‘put’, ‘keep’, ‘move’, ‘place’. This might be an artifact of the examples given to the human annotators. In any case, this word is inconsequential as the agent always lifts one of 3 available objects on one of 2 fixed objects.
- It seems like the most diversity is coming from synonyms which can probably be handled with a dictionary or wordnet rather than requiring a language model. There is also some prior work on handling synonyms (https://arxiv.org/pdf/1902.04546.pdf). I would have liked to see many more tasks and a multi-task learning model which is also able to distinguish between the task based on natural language instructions in addition to understanding object word synonyms and referring expressions. More objects would also help.
- The authors claim to tackle “more behavioural and environmental challenges than previous work”. I do not agree with this claim. It is true that this paper handles object interaction and natural language instructions in a partially observable setting, however, previous work has tackled other challenges which this work does not tackle. For example, Oh et al. 2017 generalize to new sequence of instructions, Hermann et al. 2017 and Chaplot et al. 2018 also handle compositionality and generalize to unseen instructions referring to new objects, Hermann et al. 2017 handle negation, Chaplot et al. 2018 handle instructions involving ‘largest’ or ‘smallest’ objects, Misra et al. 2018 handle more diverse natural language and so on.
- I wouldn’t call moving objects using high-level symbolic actions as ‘manipulation’. This is a whole research area in robotics involving taking low-level actions to move an object. Also, the environment used in the paper is not visually realistic in my opinion. It looks game-like and visual encoders trained in this environment are unlikely to generalize to the real world. This is fine as it is mostly irrelevant to handling natural language instructions, but authors should not claim visual realism and object manipulation in my opinion.


Comments/Questions
- I do not understand the meaning and purpose of some actions. Why are there GRAB + actions? Doesn’t the object move with the agent once it is grabbed? What is SPIN_OBJECT? Why is it needed? It seems like there is no ‘place’ action, how does the agent place the object? I am guessing when the agent stops taking GRAB+ actions. If that is the case, then wouldn’t it be easier to just have Grab and Place actions rather than 16 GRAB+ actions?
- The meaning of ‘(sub)-’ in (sub)-word is not described.
- What is the probability of typo noise in the experiments?
- Many experimental details are missing. How long was the model trained for both the tasks? How many training samples/episodes? What were the hyperparameters used for reinforcement learning? Learning rate, optimizer, discount value and so on.

Updates after author response:
I have examined the author response and additional experiments carefully. I am maintaining my score due to the following reasons:
- The authors seem to agree that the paper does not provide any 'substantial algorithmic advance'.
-  The authors argue that the number of objects in the paper is much more than prior work. However, the focus of prior work (referenced by authors) was not to tackle natural language. Since the focus of this paper is to tackle natural language, I believe the number and diversity of objects and tasks need to be much higher than 80 objects and 2 tasks used in the paper.
- "We would language referring to an even wider range of motor-behaviours, e.g. more 'verbs'" -> I do not understand what the authors are trying to say here, but if "learning the motor programmes for such concepts in an environment" is not "the focus of the present paper", I believe the focus is only handling synonyms and referring expressions for objects. In my opinion, these are relatively easier to tackle, (for example using a dictionary or wordnet) than grounding 'verbs' into sequence of actions, which limits the scope of this paper further.
- The authors still claim to tackle object manipulation and visual realism which I do not agree with. I do not believe taking high-level "GRAB" action can be called object manipulation. The objects are taken from shape net with relatively realistic shapes, but neither the appearance of objects (textures, shadows) nor the relative arrangement of objects is realistic. In my opinion, a model trained in this environment has no hope of generalizing to the real-world.
- I do not agree with the authors' argument against chance performance in the lifting task. From my experience, I believe an RL agent trained without any language input would perform at 50% if it receives a reward for lifting one of the two available objects. I do not understand why the authors chose to put only 2 objects in the environment. Why not put 5 or more objects?
- I appreciate authors' efforts towards multi-task learning results, however, tackling only 2 tasks is not convincing enough.




**Experience Assessment:**

I have published one or two papers in this area.

**Review Assessment: Checking Correctness Of Derivations And Theory:**

N/A

**Review Assessment: Checking Correctness Of Experiments:**

I assessed the sensibility of the experiments.

**Review Assessment: Thoroughness In Paper Reading:**

I read the paper thoroughly.

---

### Official Review · AnonReviewer2 · 2019-10-24
**Official Blind Review #2**

**Rating:** 3

**Review:**

This paper considers the task of instruction following where an agent navigates/interacts with a 3D environment conditioned on goals provided in natural language. While several existing approaches use synthetic language for instructions, the authors tackle this problem under the setting of noisy instructions provided by humans in natural language. For this, they use large-scale pre-trained representations (e.g. BERT) as initial parameters for representing the textual instructions. Their main result is the demonstration of transfer from agents trained using synthetic instructions to environments with more variation (e.g. synonyms) or natural instructions provided by humans on two tasks involving object manipulation.

Pros:
1. Nice application of BERT to grounded instruction following tasks
2. Good empirical results

Cons:
1. Not much technical novelty
2. Empirical experiments could use a bit more rigor in terms of disentangling the major factors that contribute to performance (e.g typo noise)



Other comments:
1. Are the BERT weights frozen or finetuned along with the rest of the model? Does the performance depend on this?
2. The typo noise (TN) seems to be a key driver of performance. Have you tried adding it to the other baselines like wordPiece Transformer?
3. What are the scores when training on the test tasks (D.O synonym, natural instructions, etc.) directly? It would be good to establish how well the transfer setup is doing compared to the best RL agent trained directly on the test scenarios.

—————
Post rebuttal update:
Thanks to the authors for their response and for updating the paper! I especially appreciate the additional experiments, but I’m still confused why the authors do not perform a clear ablation study to support their claims. It seems like the main claimed novelty of the paper is the proposed CMSA method. However, the empirical results are not convincing/rigorous enough to provide the reader information on 1) whether CMSA is a useful method (since it is used only with BERT and does not seem to affect results on its own compared to MP, SA, TN, etc.) and 2) when should one use/not use BERT and CMSA (BERT+CMSA actually does quite poorly acc. to table 5). Further, the other reviewers also pointed out concerns regarding the difficulty of the task and complexity of language used. Hence, I feel the paper still requires some revision to form a coherent story — updating my score accordingly.

**Experience Assessment:**

I have published one or two papers in this area.

**Review Assessment: Checking Correctness Of Derivations And Theory:**

N/A

**Review Assessment: Checking Correctness Of Experiments:**

I assessed the sensibility of the experiments.

**Review Assessment: Thoroughness In Paper Reading:**

I read the paper at least twice and used my best judgement in assessing the paper.

---

### Author Response · Authors · 2019-11-12
**Additional baselines, new multi-task agent and responses to principal concerns**

Thank you for your efforts in considering the paper; we're glad that overall you find some merit in the work. Moreover, your comments have allowed us to make some substantial improvements to the paper.

General improvements (detailed reviewer-specific response to follow)

Reviewer 1 called into question whether or not the results were really notable
 given that the agent is still some way from perfect on evaluation episodes, particularly with human instructions. To better appreciate exactly what the agent must do in these episodes, and thus the many sources of potential errors, we've made an (anonymous) video of the agent responding to human instructions, both succeeding and failing ( https://drive.google.com/open?id=1FM1ikS0VifdlnUACqT9HaBPpkYqWmTaP ). Note that, even for the lifting evaluation with only two objects in the room, *chance* performance is not really 50% (as it would be for a classification task), because the agent can fail either by lifting the wrong object or simply failing to lift anything. Showing that BERT representations afford robustness not just for classifiers but also in the context of behaviour and action is the fundamental novelty of this work, and we find the extent to which this is the case to be quite remarkable and unexpected (although such a reaction may of course vary depending on one's intuitions).

We understand why Reviewer 1 has raised the question of the technical/algorithmic novelty of the work. We agree that the key contribution of this work is not in the precise algorithm but in the 'recipe' for achieving more robust instruction-following by integrating several (typically 'known') components into a single agent. It seems to us that many (most?) published papers are at some level derivative of, or aggregations of, prior ideas, often applied in new ways or to new problems, and that such papers constitute many of the most impactful contributions of recent years. That said, our work is not entirely bereft of technical novelty. For instance, we are not aware of the application of cross-modal self-attention (with a temporal aspect) for mixing language and vision in a goal-driven agent. We do not consider this a substantial algorithmic advance, but it serves to illustrate how novelty in this regard (when exactly can one algorithm or architectural choice be said to be 'different' from another) is always somewhat subjective.

The main improvements to the paper are:

-- Two further baselines (Word-level transformer + TN and WordPiece transformer + TN) isolating the role of typo noise in robustness.
-- The addition of a multi-task agent trained jointly on both on the lifting and putting tasks.
-- A video illustrating the complexity of agent policies and noisiness of evaluation instructions.
--Revision of claims/clarification of environment in accordance with reviewer recommendations

---

### Author Response · Authors · 2019-11-12
**Detailed response to Reviewer 1**

Thank you for your thoughtful review. We hope that some of your concerns have been addressed in the general comments above.

On the concern that we train a separate model for both lifting and putting tasks, after reading your comments we were in fact able to train and evaluate a single agent on both tasks. We have added the full results to the paper (Tables 4 and 5). We find that "Multi-task BERT+CMSA+TN" performs only marginally worse on both lifting and putting tasks than the same architecture specialised to each individual task (typically 2-4 percentage points - and in one evaluation it actually improves). We hope this provides some evidence that our method could scale towards a single agent able to execute a wide set of instructions (indeed, the bottleneck may well not be the range of language per se but the challenge of learning realistic policies that correspond to a wide range of behaviours).

We understand your observation that there are only three movable and two fixed objects in any trial of the putting evaluation. However, selecting an incorrect object is only one way in which the agent might fail at test time if its representations are not sufficiently robust. During evaluation, both the 'putting' behaviour and object identification retain their integrity in the face of unfamiliar input. That is why we find an accuracy ~70%  in the context of entirely unfamiliar and often highly ambiguous input to be notable.

Regarding your concern about the limited number of objects/tasks in our study, we chose the present number of objects as a compromise between training burden on the agent, the computational cost of running the environment, and a desire to extend meaningfully beyond prior work. By incorporating 80 ShapeNet assets, the multi-task agent that we have just added to the paper must learn to identify (and interact with) with over ten times as many objects as in Oh et al (2017), over twice as many as in Hermann et al (2017) and over five times as many as in Jiang et al (2019). Regarding your desire to see a multi-task learning model, the multi-task BERT model that we have just trained now carries out 126 distinct `tasks' (as defined by the combination of a template language string and reward function), in 9752 different possible contexts (as defined by the other objects present in the environment) in rooms where initial positioning of objects varies continuously and randomly. We would language referring to an even wider range of motor-behaviours, e.g. more 'verbs', but learning the motor programmes for such concepts in an environment like this is a significant research challenge (and somewhat orthogonal to the focus of the present paper).

We have modified the claim to tackle "more behavioural and environmental challenges than previous work", as you point out this is somewhat subjective. Previous work tackles many of the challenges that we address here, and some challenges that we do not. The intention of Table 1 was to illustrate that the *combination* of aspects that we tackle in this work (together with fully natural language) represents a step forward in this field of research. We find this to be a reasonable claim (see next para for details of control/visual realism etc) but please check the new introduction and let us know if it is still problematic for you.

Similarly, we agree that our use of the term 'manipulation' was loose, as the term has a particular meaning in continuous control settings / robotics. We meant to distinguish our environment, where behaviours like putting require complexes of hand movement actions (grip, move-hand-up, move-hand-left, rotate-hand etc), from those such as CHALET in which object interaction involves only PICK and PUT_DOWN actions. Exactly what qualifies as visual realism is also subjective. We used the term because the ShapeNet models that we consider in the lifting task have sufficient realism to be the object of a body of recent computer vision research (see e.g. https://shapenet.cs.stanford.edu/iccv17/), particularly the growing topic of 3D scene processing.

On the action set: your are correct that an object is held by taking actions containing GRAB and otherwise dropped. Many action factorisations are possible, and this seemed the simplest that enables the agent to do interesting things with objects in the room. As you point out, another option is a GRAB action that is maintained until a DROP action is taken. In this case the GRAB+ hand movement actions may still be necessary for objects to be placed adequately on top of things. The SPIN actions allow the agent to rotate objects, which can make it less likely that they will roll off if dropped on top of the bed or tray.

Finally, we have explained the meaning of sub-word (2.1, para 2), added the probability of a typo in a given letter (1/100) (2.1, para 6) and revised Appendix C to include all details of agent training and hyperparams.

---

### Author Response · Authors · 2019-11-12
**Detailed response to Reviewer 2**

Thanks for your review. Like Reviewer 1, you raised the question of technical novelty, which we have responded to in the general comments above. We have addressed your wish for more detail in the empirical results to enable disentangling of major factors by running a couple of further conditions, "Word-level transformer + Typo Noise" and "WordPiece transformer + Typo Noise" (results in Table 4 and Table 5, and Appendix A). As would be expected, this works better with WordPiece than word-level encoding, and it is more effective on tasks involving human instructions than synonym alterations. For the WordPiece transformer on the putting task with human instructions, typo noise improves performance from 42 to 57%. The different architectures with BERT weights and typo noise score 66%, 69% and 70%, which shows that, in addition to typo noise, BERT itself contributes a significant amount to the robustness of the agent.

On the question of fine-tuning BERT, training (rather than just querying) BERT in the distributed setting of our RL algorithm requires more GPU memory than we have access to (it has many more parameters than the other components of the agent). However, as stated in section 2.1, paragraph 3, even if this were possible we believe it could cause substantial overfitting to the environment language (i.e. the benefits of BERT would be 'washed away' and overwritten by task-specific knowledge). As noted in the conclusion, overcoming this via regularization, distillation or other methods (and overcoming the engineering challenges of training a large language model with distributed RL algorithms) are interesting avenues for future research.

Re. training on human data, we did not build our pipeline to support this because we wanted to explore methods for training agents to follow instructions without involving (potentially expensive) human input in the training process (i.e. by transferring from template language, which can be generated arbitrarily in simulation). However, the question of human-in-the-loop training of agents is an interesting one, and we would certainly like to investigate this in future work.

---

### Author Response · Authors · 2019-11-12
**Answers to questions of Reviewer 3**

-- One possible explanation is that BERT's representations for these very short imperative sentences are not very good (the training environment instructions include e.g. "Lift an electric refrigerator", "Lift a sword"). It may be that such sentences are very rare in the sort of language that BERT was trained on.

-- Figure 2 compares the output of the self-attention layer of a trained BERT + SA model with the output of a self-attention layer with random weights (to show what the SA layer learns to do to BERT representations as the agent learns. We did not do this analysis with a BERT + CMSA model because the CMSA layer is conditioned on language and visual input, and it is not clear which exactly what frames of visual input should be used for the analysis. We expect the CMSA layer should operate in a similar way wrt. word representations. We have made this more clear in the text.

--Note that training performance of the BERT+CMSA model is also worse than BER+MP, to a similar degree to the evaluation performance. The lifting task is much more visually taxing than the 'putting task' for the agents (it involves a diverse space of ShapeNet models). It is possible that CMSA interferes with the agent's visual representations, in a way that inhibits the model from learning (and also results in lower test performance.

---

### Decision · Program_Chairs · 2019-12-19

**Decision:**

Reject

**Comment:**

The paper examines whether it is possible to train agents to follow synthetic instructions that perceives and modifies a 3D scene based on a first-person viewpoint, and have the trained agents follow natural language instructions provided by humans.

The paper received two weak rejects and one weak accept.  The main concerns voiced by the reviewers are:
1. Lack of variety in natural language
One of the key claims of the paper is that previous work on instruction following can only handle instructions generated from templates and cannot handle ambiguous expressions used by real people, and that the contribution of this work is that it can handle such expresssions.  However, as pointed out by R1, the language considered in this work is very simplistic in form (close to being template based) with the main variation coming from synonyms.  Even the free-form natural instructions that are collected, are done so with very specific instructions that restrict diversity of language (e.g don't use colors or other properties of the object). R1 also point out that there are prior work that handles much more diverse language.

2. Limited technical novelty and questions about how much the proposed CMSA method actually contribute

3. Overclaims and lack of precision when using terminology
There is concern that the task that is addressed is not actually that complex.  The environments are simple (with just 2 objects) and not that realistic.  Tackling 2 tasks is barely "multi-task", and commonly, "manipulation" refers to low-level grasping/picking up of objects which is not how it is used here.

While the paper has many strong elements and is mostly well written, considerable improvements still need to be made for the paper to have claims it can support.  It is currently below the bar for acceptance. The authors are encouraged to improve their paper and resubmit to an appropriate venue.